# A single-dose live-attenuated vaccine prevents Zika virus pregnancy transmission and testis damage

Chao Shan[1], Antonio E. Muruato[2], Brett W. Jagger[3], Justin Richner[3], Bruno T.D. Nunes[1,4], Daniele B.A. Medeiros[1,4], Xuping Xie[1], Jannyce G.C. Nunes[1,4], Kaitlyn M. Morabito[5], Wing-Pui Kong[5], Theodore C. Pierson[6], Alan D. Barrett[7,8,9,10], Scott C. Weaver[2,10,11,12], Shannan L. Rossi[7,8,9], Pedro F.C. Vasconcelos[4,13], Barney S. Graham[6], Michael S. Diamond[3,14,15,16] & Pei-Yong Shi[1,10,11,13,17]

Zika virus infection during pregnancy can cause congenital abnormities or fetal demise. The persistence of Zika virus in the male reproductive system poses a risk of sexual transmission. Here we demonstrate that live-attenuated Zika virus vaccine candidates containing deletions in the 3′ untranslated region of the Zika virus genome (ZIKV-3′UTR-LAV) prevent viral transmission during pregnancy and testis damage in mice, as well as infection of nonhuman primates. After a single-dose vaccination, pregnant mice challenged with Zika virus at embryonic day 6 and evaluated at embryonic day 13 show markedly diminished levels of viral RNA in maternal, placental, and fetal tissues. Vaccinated male mice challenged with Zika virus were protected against testis infection, injury, and oligospermia. A single immunization of rhesus macaques elicited a rapid and robust antibody response, conferring complete protection upon challenge. Furthermore, the ZIKV-3′UTR-LAV vaccine candidates have a desirable safety profile. These results suggest that further development of ZIKV-3′UTR-LAV is warranted for humans.

[1] Department of Biochemistry & Molecular Biology, University of Texas Medical Branch, Galveston, TX 77555, USA. [2] Department of Microbiology & Immunology, University of Texas Medical Branch, Galveston, TX 77555, USA. [3] Department of Medicine, Washington University School of Medicine, St. Louis, MO 63110, USA. [4] Department of Arbovirology and Hemorrhagic Fevers, Evandro Chagas Institute, Ministry of Health, 67030-000 Ananindeua, Brazil. [5] Vaccine Research Center, National Institute of Allergy and Infectious Diseases, National Institutes of Health, Bethesda, MD 20892, USA. [6] Viral Pathogenesis Section, Laboratory of Viral Diseases, National Institute of Allergy and Infectious Disease, National Institutes of Health, Bethesda, MD 20892, USA. [7] Institute for Human Infections & Immunity, University of Texas Medical Branch, Galveston, TX 77555, USA. [8] Department of Pathology, University of Texas Medical Branch, Galveston, TX 77555, USA. [9] Center for Biodefense & Emerging Infectious Diseases, University of Texas Medical Branch, Galveston, TX 77555, USA. [10] Sealy Center for Vaccine Development, University of Texas Medical Branch, Galveston, TX 77555, USA. [11] Institute for Translational Science, University of Texas Medical Branch, Galveston, TX 77555, USA. [12] Sealy Center for Structural Biology & Molecular Biophysics, University of Texas Medical Branch, Galveston, TX 77555, USA. [13] Department of Pathology, Pará State University, 67087-670 Belém, Brazil. [14] Department of Molecular Microbiology, Washington University School of Medicine, St. Louis, MO 63110, USA. [15] Department of Pathology & Immunology, Washington University School of Medicine, St. Louis, MO 63110, USA. [16] The Andrew M. and Jane M. Bursky Center for Human Immunology and Immunotherapy Programs, Washington University School of Medicine, St. Louis, MO 63110, USA. [17] Department of Phamarcology & Toxicology, University of Texas Medical Branch, Galveston, TX 77555, USA. Correspondence and requests for materials should be addressed to B.S.G. (email: bgraham@nih.gov) or to M.S.D. (email: diamond@wusm.wustl.edu) or to P.-Y.S. (email: peshi@utmb.edu)

Zika virus (ZIKV) belongs to the Flavivirus genus in the *Flaviviridae* family, and is closely related to other globally important flaviviruses, including dengue, yellow fever, Japanese encephalitis, West Nile, and tick-borne encephalitis viruses[1]. Flaviviruses have a single-stranded, positive-sense RNA genome consisting of a 5′ untranslated region (UTR), a single open reading frame (ORF), and a 3′UTR. The single ORF encodes three structural (C-prM/M-E) and seven nonstructural (NS1-NS2A-NS2B-NS3-NS4A-NS4B-NS5) proteins. ZIKV is transmitted primarily by peridomestic *Aedes* mosquitoes, but also can be acquired through sexual, vertical, and blood transfusion routes[2, 3]. Although only about 20% of infected individuals develop clinical sign or symptoms, ZIKV infection in some adults is associated with the development of Guillain–Barré syndrome[4]. The most devastating manifestation of ZIKV infection is the array of congenital abnormalities in the fetuses and infants of women infected while pregnant, including microcephaly, craniofacial disproportion, spasticity, seizures, ocular abnormalities, cerebral calcification, and miscarriage[5, 6]. Longitudinal studies in humans in Brazil showed that among live infants born to symptomatic ZIKV-positive women, 42% had grossly abnormal clinical or brain imaging findings[5]. Furthermore, ZIKV infection can persist in the male reproductive tract, as infectious virus and viral RNA

have been detected in semen up to 69 and 188 days after symptom onset, respectively[7–9]; infected males can transmit virus to sexual partners during this period of persistent infection[10]. Studies in mice have shown that ZIKV persistence in the testis causes damage to the seminiferous tubules, testicular atrophy, oligospermia, and reduced rates of fertility[11–13]. A number of promising ZIKV vaccine platforms have been developed, including inactivated, subunit (prM-E proteins expressed from DNA, RNA, or viral vectors), and live-attenuated vaccines[14–21], several of which have entered phase I-II clinical trials[22, 23]. Among these vaccine candidates, the prM-E RNA vaccine and live-attenuated vaccine (lacking NS1 glycosylations) were recently shown to protect against vertical transmission in a mouse pregnancy model[24]. In addition, the DNA vaccine expressing prM-E proteins was recently shown to protect mice against ZIKV-induced damages to testes[25].

Here we show that live-attenuated ZIKV vaccine candidates containing deletions in the 3′UTR of the ZIKV genome (ZIKV-3′UTR-LAV) prevent viral transmission during pregnancy and testis damage in mice, as well as infection of nonhuman primates (NHPs). We also demonstrate a desirable safety profile of the vaccine candidates. Our data suggest that ZIKV-3′UTR-LAV merits further development for humans.

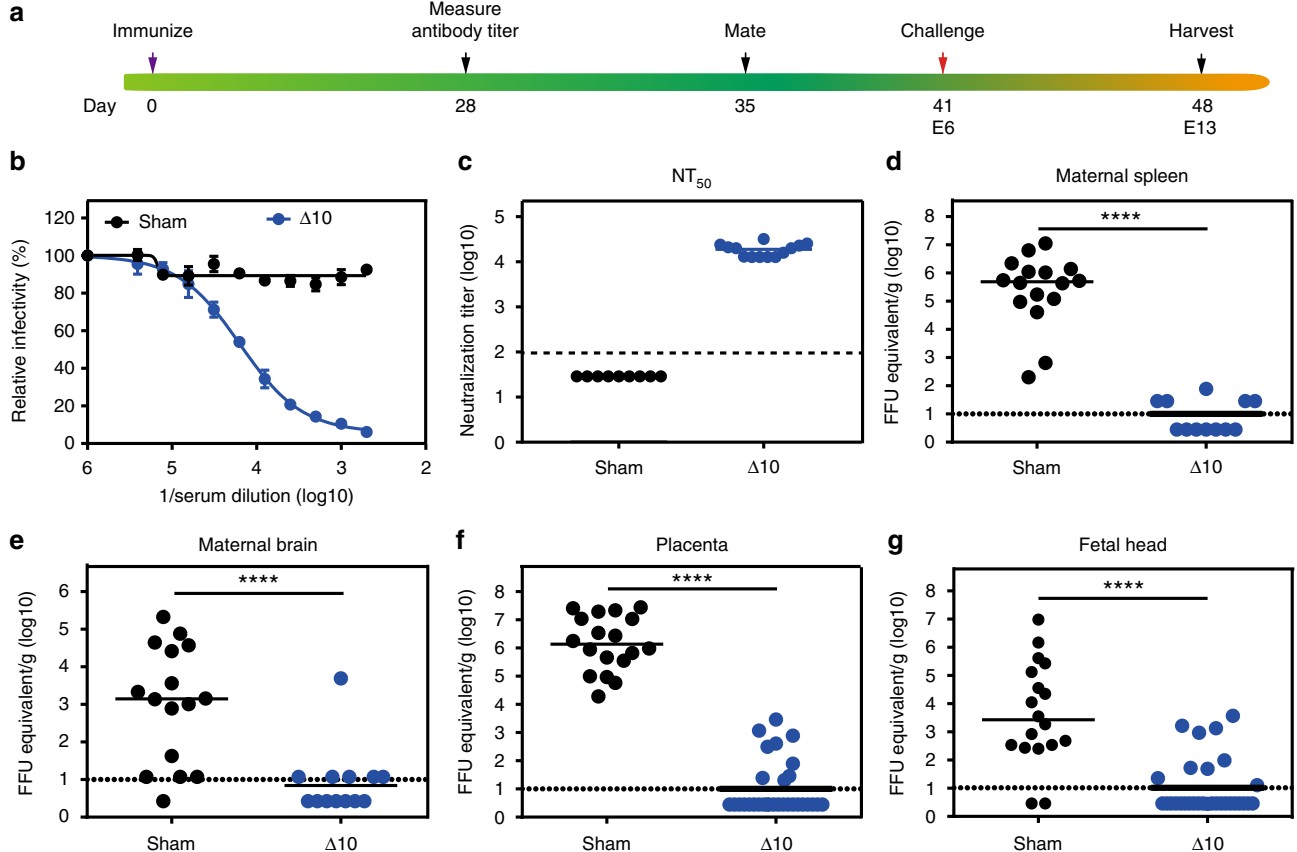

**Fig. 1** ZIKV-3′UTR-Δ10-LAV protects pregnant C57BL/6 mice and their developing fetuses. **a** Scheme of immunization of wild-type (WT) C57BL/6 female mice with $10^5$ FFU of ZIKV-3′UTR-Δ10-LAV (Δ10; n = 12) or PBS sham (n = 16). **b** Serum was collected at day 28 post immunization and analyzed for neutralizing activity using an mCherry infectious ZIKV. Representative neutralization curves are shown. *Error bars* denote the SD of duplicate technical replicates. **c** $NT_{50}$ values of neutralizing antibodies were measured for individual animals. The *dashed lines* indicate the limit of detection (LOD) of the assay. **d–g** At day 35 post immunization, vaccinated female mice were mated with WT C57BL/6 males. A subset of the female mice developed vaginal plugs. Pregnant mice (n = 8 pooled from two independent experiments) were administered 2 mg of anti-Ifnar1 blocking antibody on E5, and 1 day later (E6), challenged with $10^5$ FFU of a pathogenic, mouse adapted ZIKV Dakar 41519 strain. On E13, animals were euthanized; maternal spleen (**d**), maternal brain (**e**), placenta (**f**), and fetal heads (**g**) were harvested and quantified for viral RNA levels. Median viral RNA levels are indicated for each group. *Asterisks* indicate significant differences (Mann–Whitney test: ****P-value < 0.0001). All negative samples are plotted at the half value of LOD. The results in the Figure are pooled from two independent experiments

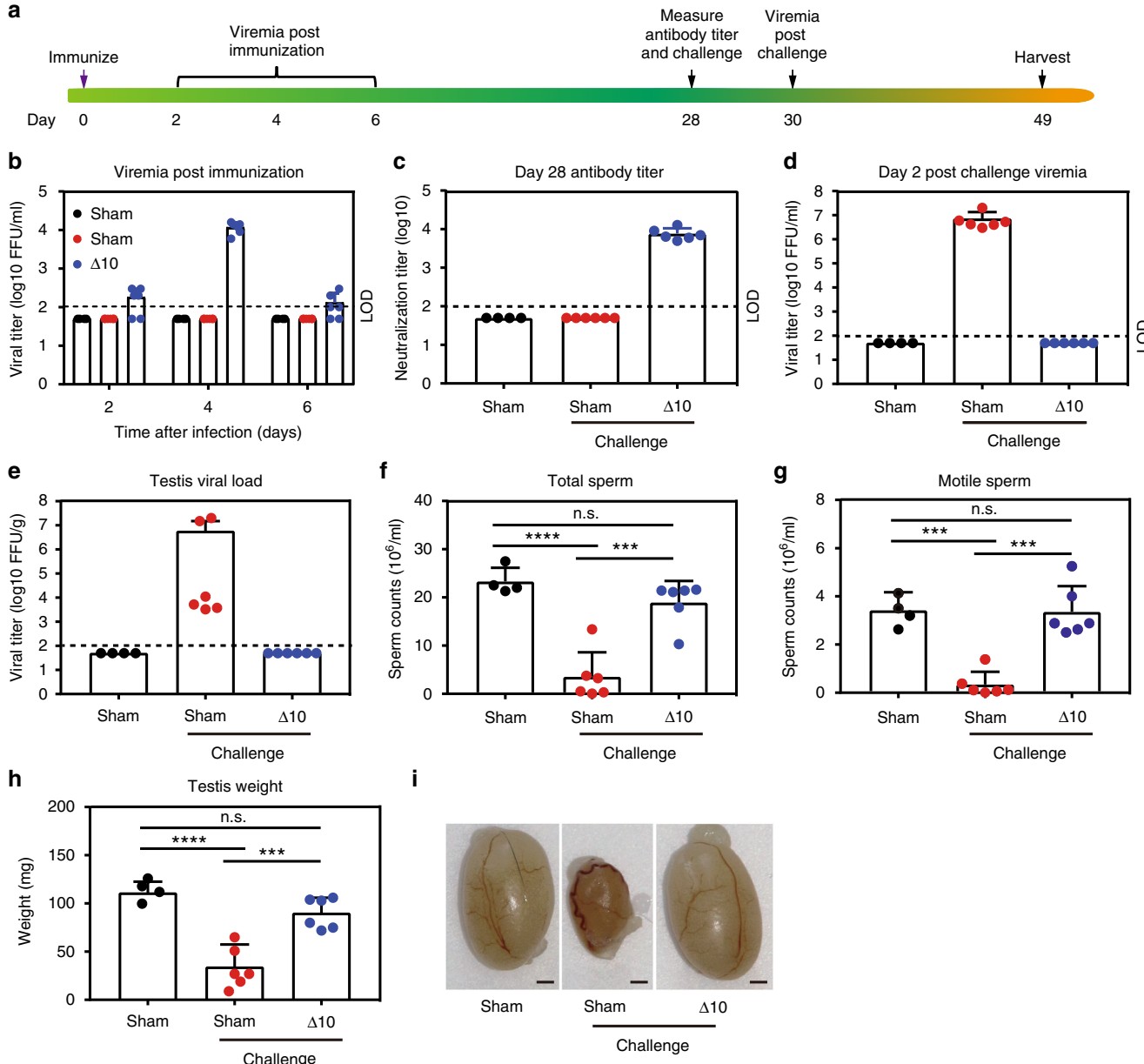

**Fig. 2** ZIKV-3′UTR-Δ10-LAV protects young A129 male mice against testis infection and injury. **a** Scheme of immunization of 3-week-old A129 male mice with 10$^4$ FFU of ZIKV-3′UTR-Δ10-LAV (Δ10; $n=6$) or PBS sham ($n=4$ or 6). At day 28 post immunization, mice were measured for neutralization antibody titers. On the same day, mice from one sham group and mice from Δ10-immunized group were challenged with 10$^6$ FFU of ZIKV-PRVABC59, and viremia was measured at day 2 post challenge (day 30 post immunization). At day 49 post immunization, mice were analyzed for sperm counts and viral load in the testis. **b** Viremia after immunization with Δ10 vaccine candidate. **c** NT$_{50}$ values of antibody neutralization at day 28 post immunization were measured for individual animals in each group. The *dashed lines* indicate the limit of detection (LOD) of the assay. **d** Viremia at day 2 post challenge (day 30 post immunization) with ZIKV PRVABC59. **e** Viral load in the testis at day 21 post challenge (day 49 post immunization). **f, g** Total (**f**) and motile (**g**) sperm counts at day 21 post challenge. **h, i** Testis weight (**h**) and representative images of testis (**i**) from animals from sham, sham with challenge, and Δ10-immunized and challenged groups at day 21 post challenge. *Scale bar*, 1 mm. Asterisks indicate significant differences (one-way ANOVA: ****$P$-value < 0.0001; ***$P$-value < 0.001). Nonsignificant (n.s.), $P$-value > 0.05. All negative samples are plotted at the half value of LOD. *Error bars* represent SDs

## Results

**Prevention of vertical transmission in pregnant mice.** We recently developed a live-attenuated vaccine candidate containing a 10-nucleotide deletion in the 3′UTR of ZIKV genome (ZIKV-3′UTR-Δ10-LAV); this vaccine conferred nearly sterilizing immunity after a single-dose immunization and showed an excellent safety profile in mice[19]. To test the ability of this vaccine candidate to prevent in utero transmission, we subcutaneously inoculated 10$^5$ focus-forming units (FFU) of ZIKV-3′UTR-Δ10-LAV or Phosphate Buffered Saline (PBS-

sham) into 8-week-old wild-type (WT) C57BL/6 female mice (Fig. 1a). Because mice are not a native host for ZIKV due in part to a species-dependent lack of antagonism of type I interferon (IFN) signaling[26, 27], we administered 0.5 mg of anti-IFN alpha/beta receptor 1 (anti-Ifnar1) blocking antibody to female mice 1 day prior to vaccination to facilitate transient replication of ZIKV-3′UTR-Δ10-LAV[28]. At day 28 post-vaccination, the animals were phlebotomized and serum was analyzed for neutralizing antibody (Fig. 1b); all ZIKV-3′UTR-Δ10-LAV-immunized mice developed high neutralizing antibody titers of

18,900 ± 5900 (mean ± SD; $n = 13$), whereas, as expected, the PBS-immunized animals did not develop detectable neutralizing antibodies (Fig. 1c). At day 35 post vaccination, the immunized females were mated with 12-week-old WT C57BL/6 male mice and monitored for vaginal plugs (Fig. 1a).

At embryonic day 6 (E6), pregnant mice were challenged subcutaneously with $10^5$ FFU of a mouse-adapted, pathogenic ZIKV African strain Dakar 41519[29]; to facilitate ZIKV-Dakar dissemination to the maternal decidua and fetal placenta, the pregnant mice were administered 2 mg of anti-Ifnar1 antibody at

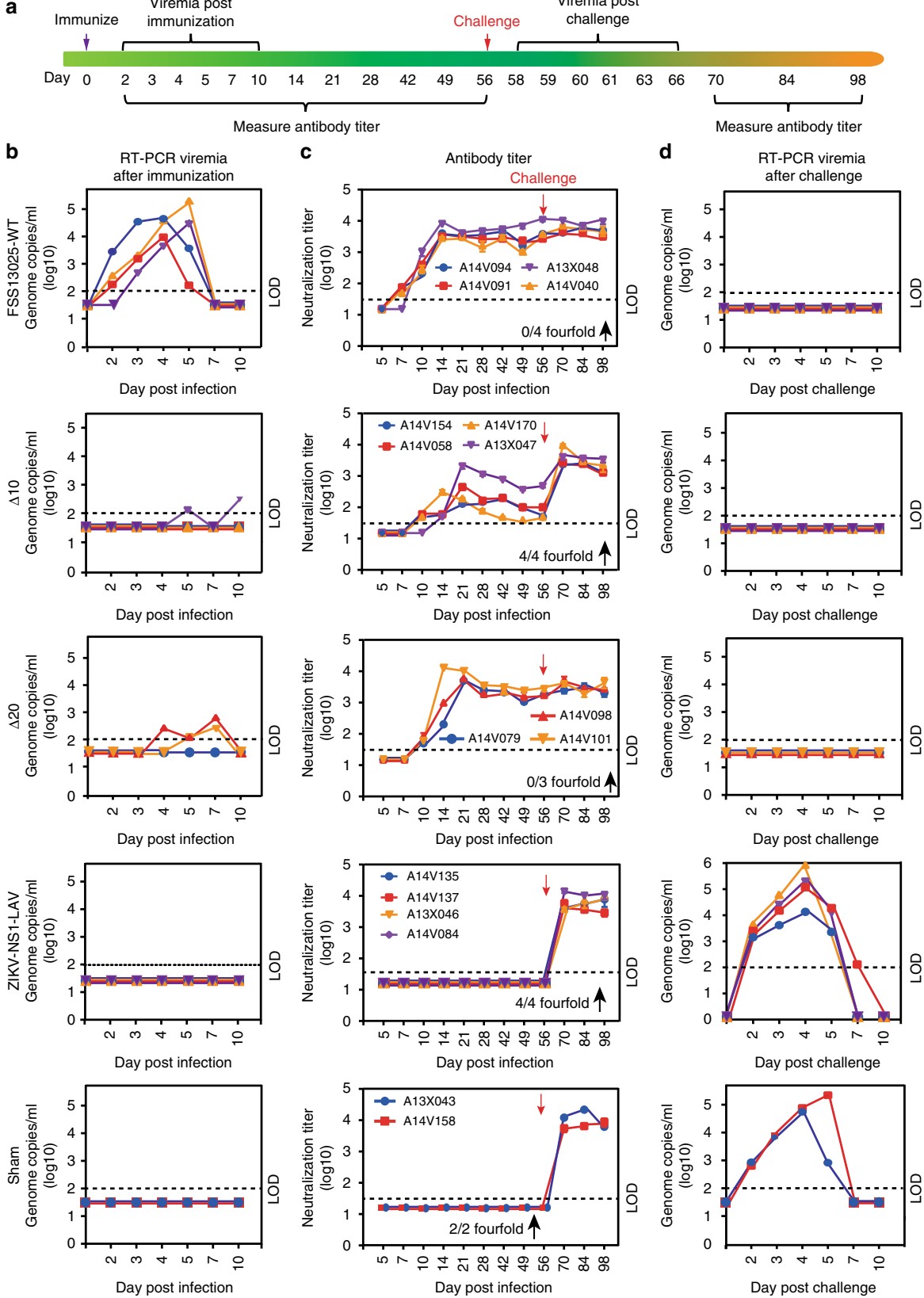

E5, 1 day before the challenge. At E13, maternal and fetal organs were harvested and measured for viral load.

After a single immunization, ZIKV-3′UTR-Δ10-LAV reduced median viral loads in the maternal spleen and brain by ~49,000-fold and ~120-fold, respectively (Fig. 1d, e). Placentas and fetal heads from the vaccinated dams showed 138,000-fold and 260-fold decreases in viral RNA loads, respectively, when compared with the PBS-immunized dams (Fig. 1f, g). Notably, 21 of 30 (70%) placentas and 21 of 30 (70%) fetal heads from ZIKV-3′UTR-Δ10-LAV-immunized dams had viral RNA loads at or below the detection limit; no infectious virus was recovered by focus-forming assay from the placentas or fetal heads from the vaccinated dams (Supplementary Fig. 1a, b). In addition, the immune correlate between the NT$_{50}$ values and the levels of infectious viruses detected in the placenta revealed an expected inverse relationship between neutralizing titers and levels of ZIKV particles in placenta (Supplementary Fig. 1c). Collectively, the data suggest that ZIKV-3′UTR-Δ10-LAV protects maternal organs from infection, prevents viral transmission to the fetus during an early stage of pregnancy, and limits fetal virus replication.

**Protection of ZIKV-induced damages to testes.** We examined the ability of ZIKV-3′UTR-Δ10-LAV to prevent testis infection and injury in *Ifnar1*$^{-/-}$ A129 mice (Fig. 2a). Since male mice reach sexual maturity at 8 weeks of age, we tested the vaccine efficacy in two age groups of mice: 3-week-old young males; and 15-week-old adult males. Three-week-old young A129 male mice were vaccinated with a single-dose of 10$^4$ FFU of ZIKV-3′UTR-Δ10-LAV or PBS sham. The ZIKV-3′UTR-Δ10-LAV generated a peak viremia of $1.2 \pm 0.34 \times 10^4$ FFU (n = 6) at day 4 post vaccination (Fig. 2b). At day 28 post vaccination, ZIKV-3′UTR-Δ10-LAV had induced robust neutralizing antibody titers of $7700 \pm 2600$ (n = 6; Fig. 2c). At the same day, the animals were challenged intraperitoneally with 10$^6$ FFU of an epidemic ZIKV strain from Puerto Rico (PRVABC59). No viremia was detected from the ZIKV-3′UTR-Δ10-LAV-vaccinated mice after challenge, whereas the sham-vaccinated animals sustained a mean peak viremia of $7.1 \pm 5.9 \times 10^6$ FFU/ml (n = 6) at day 2 post challenge (Fig. 2d). At day 21 post challenge, viral burden in the testis of PBS-immunized mice reached $5.8 \pm 8.4 \times 10^6$ FFU/g (n = 6), whereas no infectious virus was detected in the testes of ZIKV-3′UTR-Δ10-LAV-immunized mice (Fig. 2e). Both total and motile sperm counts from the ZIKV-3′UTR-Δ10-LAV-immunized mice were equivalent to those from age-matched unvaccinated, unchallenged healthy male mice. In contrast, the PBS-immunized, ZIKV-challenged mice showed 85% and 90% reductions for total and motile sperm counts, respectively, at day 21 post infection (Fig. 2f, g). Consistent with these data, the testis weight and size from the PBS-immunized, ZIKV-challenged mice were reduced, whereas no such reduction was observed in the ZIKV-3′UTR-Δ10-LAV-immunized, ZIKV-challenged animals (Fig. 2h, i).

In 15-week-old adult A129 male mice, vaccination with ZIKV-3′UTR-Δ10-LAV also protected against testis infection, injury, and oligospermia (Supplementary Fig. 2a–g). However, challenge of the PBS-vaccinated adult males with ZIKV PRVABC59 did not reduce the testis weight and size (Supplementary Fig. 2h, i) as much as that observed in the corresponding sham-vaccinated, young males (Fig. 2h, i), suggesting an age-dependent testis pathology. Collectively, the results indicate that a single-dose immunization of ZIKV-3′UTR-Δ10-LAV protects the testis from infection and injury in male mice.

**Efficacy in NHPs.** To determine whether the efficacy in mice extends to NHPs, we evaluated the viremia, immunogenicity, and potency of ZIKV-3′UTR-Δ10-LAV in rhesus macaques (RMs; Fig. 3a). First, we assessed the level of attenuation of ZIKV-3′UTR-Δ10-LAV in RMs by comparing viremia to the parental WT virus. After subcutaneous inoculation with 10$^3$ FFU, WT ZIKV (2010 Cambodian strain FSS 13025) produced high levels of viremia in each of the four inoculated RMs, with a mean peak viremia of $9.6 \times 10^4$ and $2.88 \times 10^5$ genome copies/ml at days 4 and 5 post infection, respectively (Fig. 3b, *top panel*). In contrast, only one of the four ZIKV-3′UTR-Δ10-LAV-inoculated RMs exhibited viremia, and this level was just above the limit of detection of our quantitative reverse transcriptase-PCR (qRT-PCR) assay (Fig. 3b, *second panel*); thus, ZIKV-3′UTR-Δ10-LAV is highly attenuated in RMs. We next evaluated the immunogenicity of ZIKV-3′UTR-Δ10-LAV by measuring neutralizing antibodies from serum at days 5–98 post immunization. Neutralizing antibodies elicited by WT ZIKV were detectable at day 7–10, peaked at 1/1000–1/10,000 at day 14, and plateaued thereafter (Fig. 3c, *top panel*). Compared with the WT ZIKV infection, ZIKV-3′UTR-Δ10-LAV-inoculated animals showed slightly delayed production and lower levels of neutralizing antibody, with titers of ~1/100 at days 21–56 (Fig. 3c, *second panel*). At day 56 post immunization, all RMs were challenged with 10$^3$ FFU of the epidemic ZIKV strain PRVABC59. Notably, no viremia was detected upon challenge in any of the RMs that were pre-infected with WT ZIKV or vaccinated with the ZIKV-3′UTR-Δ10-LAV (Fig. 3d, *top two panels*). As controls, two PBS-inoculated RMs were challenged in parallel and high levels of viremia were measured (Fig. 3d, *bottom panel*). In addition to analysis of viral RNA by qRT-PCR, we also performed focus-forming assays to measure infectious virus; infectious ZIKV was detected only in naive RMs challenged with WT ZIKV (Supplementary Fig. 3).

To examine whether the ZIKV challenge resulted in boosted immune responses, we measured neutralizing activity post challenge. No increase in neutralizing activity was observed in the WT ZIKV-vaccinated RMs after challenge (Fig. 3c, *top panel*), indicating that the initial infection likely conferred sterilizing immunity. In contrast, the neutralizing antibody titers rose after challenge from ~1/100 to 1/1000–1/10,000 in the ZIKV-3′UTR-Δ10-LAV-immunized animals (Fig. 3c, *second panel*), demonstrating an anamnestic response and suggesting a low level of infection after challenge that was not detectable by qRT-PCR of serum. As expected, the PBS-inoculated control RMs increased their neutralizing titers to ~1/10,000 after challenge (Fig. 3c, *bottom panel*).

**Fig. 3** ZIKV-3′UTR-Δ10-LAV and ZIKV-3′UTR-Δ20-LAV protect rhesus macaques (RMs) from ZIKV infection. **a** Scheme of immunization of RMs with 10$^3$ FFU of WT ZIKV strain FSS13025 (n = 4), ZIKV-3′UTR-Δ10-LAV (Δ10; n = 4), ZIKV-3′UTR-Δ20-LAV (Δ20; n = 3), ZIKV-NS1-LAV or PBS sham (n = 2) via the subcutaneous route. **b** Viremia was measured at days 2, 3, 4, 5, 7, and 10 post immunization by qRT-PCR. Each *colored line* represents data from a different animal in each group. The *dashed line* indicates the limit of detection (LOD) of the assay. **c** Pre- and post-challenge antibody neutralization titers. On various days post immunization, sera were measured for neutralizing titers using an mCherry ZIKV infection assay. *Red arrows* indicate challenge with 10$^3$ FFU of epidemic ZIKV strain PRVABC59 via the subcutaneous route at day 56 post immunization. The number of animals whose antibody neutralization titers increased by ≥ 4-fold after challenge is indicated by symbol "↑" for each experimental group. **d** Post-challenge viremia. Viremia was measured by qRT-PCR at days 2, 3, 4, 5, 7, and 10 post challenge. All negative samples are plotted at the half value of LOD. *Error bars* represent SDs

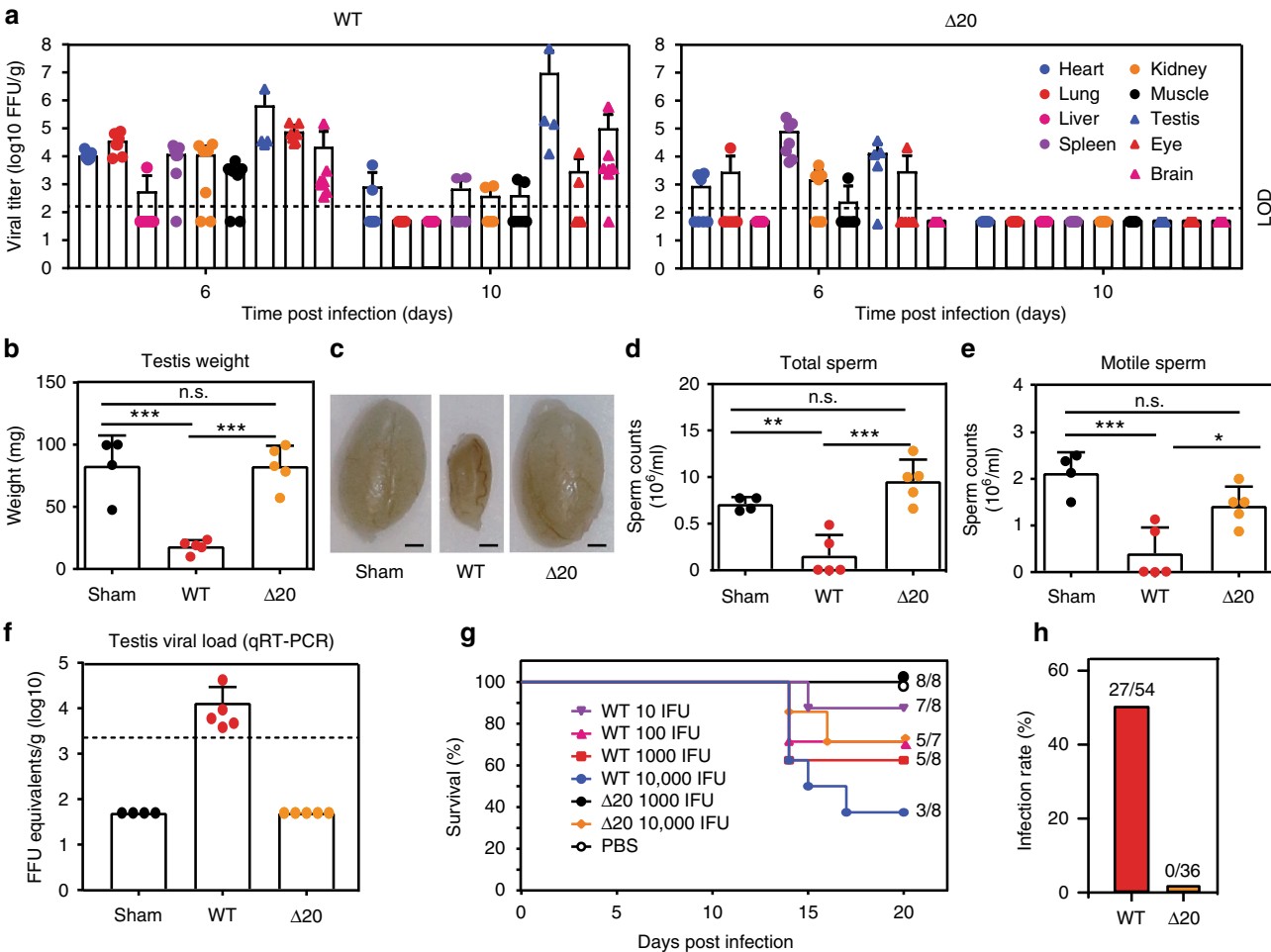

**Fig. 4** Safety evaluation of ZIKV-3′UTR-Δ20-LAV (Δ20) vaccine candidate. **a** Viral loads in organs of infected A129 mice. Three-week-old A129 mice (n = 7) were subcutaneously immunized with $10^3$ FFU of WT ZIKV FSS13025 (*left panel*) and its derivative Δ20 vaccine candidate (*right panel*). Organs from infected mice were collected and homogenized at days 6 and 10 post infection. The amounts of viruses were quantified on Vero cells using a focus-forming assay. The mean results from seven animals are presented. Bars denote standard errors. The *dashed lines* indicate the limit of detection (LOD) of the assay. **b**–**f** Effect of Δ20 vaccination on the testis. Three-week-old A129 mice (n = 5) were subcutaneously infected with $1 \times 10^3$ FFU of WT ZIKV FSS13025 or Δ20 vaccine candidate. At day 28 post infection, animals from each group were analyzed for testis weight (**b**), testis size (**c**), total sperm counts (**d**), motile sperm counts (**e**), and viral RNA load (**f**). *Scale bar*, 1 mm. **g** Comparison of neurovirulence of WT ZIKV FSS13025 and Δ20 vaccine candidate in outbred CD-1 mice. One-day-old CD-1 mice (n = 7–8/group) injected intracranially with 10–$10^4$ FFU of WT ZIKV or $10^3$–$10^4$ FFU of Δ20 vaccine candidate. Surviving mice and total infected animals are indicated. **h** Analysis of vector competency. *Aedes aegypti* were fed on artificial blood-meals spiked with $10^6$ FFU/ml of WT ZIKV FSS13025 or Δ20 vaccine virus. At day 7 post feeding, individual engorged mosquitoes were assayed for infection by immunostaining of viral protein expression on inoculated Vero cells. The number of infected mosquitoes and total number of engorged mosquitoes are indicated. *Asterisks* indicate significant differences (one-way ANOVA: ***P-value < 0.001; **P-value < 0.01; *P-value < 0.05). Nonsignificant (n.s.) with P-value > 0.05. All negative samples are plotted at the half value of LOD. *Error bars* represent SDs

Since ZIKV-3′UTR-Δ10-LAV did not elicit sterilizing immunity in RMs, we evaluated whether a second live-attenuated vaccine candidate ZIKV-3′UTR-Δ20-LAV[19], which contains a 20-nucleotide deletion in the 3′UTR, could induce a stronger immune response. ZIKV-3′UTR-Δ20-LAV was shown previously, and paradoxically, to be less attenuated than ZIKV-3′UTR-Δ10-LAV in A129 mice, most likely because it is less sensitive to type I IFN inhibition compared to ZIKV-3′UTR-Δ10-LAV[19]. After subcutaneous inoculation of $10^3$ FFU of ZIKV-3′UTR-Δ20-LAV, two of the three RMs had low, but detectable viremia (Fig. 3b, *third panel*). The immunized animals rapidly produced neutralizing antibodies by day 10, with inhibitory titers plateauing at 1/1000–1/10,000 by days 14–21 (Fig. 3c, *third panel*). After challenge with $10^3$ FFU of ZIKV PRVABC59 at day 56, viremia was not detected by qRT-PCR (Fig. 3d, *third panel*) and no rise in neutralizing antibody titers was observed (Fig. 3c,

*third panel*) in the ZIKV-3′UTR-Δ20-LAV-immunized animals. Although low number of animals were used for each vaccine candidate, the results suggest that a single-dose vaccination of ZIKV-3′UTR-Δ20-LAV induces sterilizing immunity in NHPs (i.e., no detectable viremia and no increase of neutralizing antibody titer after challenge).

We also evaluated another live-attenuated ZIKV vaccine candidate encoding an NS1 without glycosylation (ZIKV-NS1-LAV) in RMs. ZIKV-NS1-LAV was recently shown to prevent in utero transmission in a mouse pregnancy model[24]. After subcutaneous immunization of four RMs with $10^3$ FFU of ZIKV-NS1-LAV, none of the animals showed any detectable viral RNA (Fig. 3b, *fourth panel*). Back titering of the ZIKV-NS1-LAV inoculum using a focus-forming assay confirmed the infectivity of viral stock with the expected infectious titer. Unexpectedly, the immunization did not ellicit any neutralizing activity

(Fig. 3c, *fourth panel*). After challenge with $10^3$ FFU of ZIKV PRVABC59 at day 56, all four animals displayed robust viremia (Fig. 3d, *fourth panel*) and generated neutralizing antibody titers (Fig. 3c, *fourth panel*). The results indicate that ZIKV-NS1-LAV is incapable of replicating and triggering antibody response in RMs.

**Testis protection and safety analysis of ZIKV-3′UTR-Δ20-LAV.** Because of the highly desirable sterilizing immunity induced by ZIKV-3′UTR-Δ20-LAV in RMs, we further tested its efficacy and safety. Similar to ZIKV-3′UTR-Δ10-LAV, immunization of male A129 mice with $10^3$ FFU of ZIKV-3′UTR-Δ20-LAV completely prevented viral infection and testis injury after challenge with ZIKV PRVABC59, as determined by a lack of detectable viremia post challenge, the absence of oligospermia, and no decrease in testis weight and size (Supplementary Fig. 4). Next, five sets of experiments were performed to characterize the safety of ZIKV-3′UTR-Δ20-LAV. First, we measured the organ viral loads after subcutaneous inoculation of A129 mice with $10^3$ FFU of ZIKV-3′UTR-Δ20-LAV or parental WT ZIKV (Fig. 4a). At day 6 post infection, WT ZIKV-infected mice exhibited high viral loads in all tested organs, whereas no infectious virus was detected ($\leq 10^2$ FFU/ml) in liver or brain from the ZIKV-3′UTR-Δ20-LAV-infected mice, with other organs (except spleen) exhibiting lower levels of the vaccine virus than those of the WT-infected animals. At day 10 post infection, WT ZIKV-infected mice retained viral loads in the heart, spleen, kidney, muscle, testis, eye, and brain, whereas no organs from the ZIKV-3′UTR-Δ20-LAV-infected mice had any detectable virus. Second, we examined the potential adverse effect of ZIKV-3′UTR-Δ20-LAV on the testis in 3-week-old A129 mice. As expected, at day 21 post infection, WT ZIKV infection reduced testis weight and size (Fig. 4b, c), lowered total and motile sperm counts (Fig. 4d, e), and resulted in viral RNA in the shrunken testis (Fig. 4f). In contrast, ZIKV-3′UTR-Δ20-LAV did not affect sperm counts or testis weight and size (Fig. 4b–e), with no detectable viral RNA in the testes (Fig. 4f). Third, we evaluated the neurovirulence of ZIKV-3′UTR-Δ20-LAV through intracranial inoculation of 1-day-old outbred, immunocompetent CD-1 mice (Fig. 4g). As reported previously[19], neonates succumbed to WT ZIKV infection; even a dose of only 10 FFU resulted in 13% mortality (Fig. 4g). In contrast, no mortality was observed in mice that were inoculated with $10^3$ FFU of ZIKV-3′UTR-Δ20-LAV; however, infection with $10^4$ FFU of ZIKV-3′UTR-Δ20-LAV resulted in a mortality rate of 29%. Fourth, we tested if the vaccine candidate could infect *Aedes aegypti* mosquitoes, the main vector of ZIKV[30], [31]. After feeding on artificial blood-meals containing $10^6$ FFU/ml of ZIKV-3′UTR-Δ20-LAV or WT ZIKV, 50% of the engorged mosquitoes were infected by WT ZIKV, whereas no mosquitoes were infected by ZIKV-3′UTR-Δ20-LAV (Fig. 4h). Finally, we tested the stability of ZIKV-3′UTR-Δ20-LAV in cell culture. After continuous culture of ZIKV-3′UTR-Δ20-LAV on Vero cells (an approved cell line for vaccine production[32]) for five rounds, all recovered P5 viruses (derived from three independent experiments) retained the 20-nucleotide deletion. However, the P5 viruses accumulated additional mutations in the E- and NS1-encoding genes (Supplementary Table 1), which may represent Vero-cell-adaptive mutation(s) or compensatory mutation(s) to 3′UTR deletion. Further passaging of the viruses to P10 did not change the 20-nucleotide deletion, indicating that the deletion is stable in cell culture. Moreover, we passaged ZIKV-3′UTR-Δ20-LAV in A129 mice for three rounds (3 days/round); all recovered viruses retained the 20-nucleotide deletion, further suggesting the stability of the mutant virus. Taken together, these results demonstrate an excellent safety profile of ZIKV-3′UTR-Δ20-

LAV, including limited, transient viral loads in mouse organs, no adverse effect on testicular function, decreased neurovirulence, incompetency to infect mosquitoes, and good stability.

## Discussion

Our results showed that a single immunization of ZIKV-3′UTR-Δ10-LAV prevented maternal-to-fetal transmission early during pregnancy in C57BL/6 mice. Although no infectious challenge virus was detected, very low levels of viral RNA were recovered from ~30% of placentas and fetal heads from the vaccinated dams after challenge; these breakthrough viral RNAs might derive from stable antibody–virus complexes, which can last for several days in vivo[33]. The clinical implications of such breakthrough noninfectious viral RNA remain to be determined. In male A129 mice, a single-dose immunization of either ZIKV-3′UTR-Δ10-LAV or ZIKV-3′UTR-Δ20-LAV prevented testis infection and injury after challenge, indicating an additional benefit of vaccination to protect the male reproductive system. Notably, unprotected young A129 mice (3 or 7 weeks old when infected) infected with ZIKV developed smaller testes, whereas adult mice (19 weeks old when infected) did not, suggesting that ZIKV infection might cause more severe reproductive damage in younger males. The clinical relevance of this observation remains to be verified in NHPs and in humans. In NHPs, a single-dose vaccination with ZIKV-3′UTR-Δ10-LAV or ZIKV-3′UTR-Δ20-LAV induced sufficient immune responses to prevent viremia, with the ZIKV-3′UTR-Δ20-LAV eliciting greater immunogenicity, as reflected by its ability to induce sterilizing immunity against challenge. One limitation of the current NHP results is the low number of animals used for each vaccine candidates ($n = 3$–4). ZIKV vaccine-induced sterilizing immunity might be critical for protection against congenital abnormalities in humans.

Live-attenuated vaccines generally have the advantage of single dose, rapid induction of durable immunity. Since ZIKV is endemic primarily in low-income countries, a vaccine with single-dose efficacy is of practical importance, particularly when controlling an explosive outbreak or immunizing populations in remote areas where multiple doses and periodic boosting will be challenging[34]. Thus, live-attenuated vaccines may be useful for immunizing populations living in and traveling to ZIKV-endemic areas. Besides our ZIKV-3′UTR-LAV, a single-dose immunization with nucleoside-modified mRNA expressing ZIKV prM-E (50 μg)[18] or a recombinant rhesus adenovirus serotype 52 vector expressing ZIKV prM-E ($10^{11}$ viral particles)[16] was also shown to rapidly elicit antibody response and prevent viremia in NHPs; whether these two vaccines achieved sterilizing immunity was not determined. All other vaccine platforms, including inactivated vaccine and prM-E DNA vaccine, need two immunizations to elicit robust antibody response for viremia protection in NHPs[15], [16]. It is conceivable that in immunocompromised individuals and pregnant women, vaccination with live-attenuated virus may be contraindicated to avoid potential adverse risks. These individuals could be protected using inactivated, subunit, or gene-based replication-defective vaccines. Therefore, there is a need to develop multiple vaccine platforms in parallel to provide complementary options for preventing and controlling ZIKV infection and disease.

## Methods

**Experimental systems and measurements.** Various in vivo systems were employed to characterize the safety and efficacy of vaccine candidates, including C57BL/6J mouse pregnancy, A129 mouse testis protection, viral loading in A129 mouse organ, CD-1 mouse neurovirulence, and RMs efficacy (see details below). Since the protocols for each of these experimental systems have been previously established (e.g., inoculum dose, infection route, challenge dose, and end-point measurement), we did not attempt to change these optimized parameters when testing our vaccine candidates. Therefore, different inoculum doses, challenge

doses, and end-point measurements (e.g., qRT-PCR to measure viral RNA and focus-forming assay to measure infection virus) were used in different in vivo systems according to the established protocols. The detailed information is described below and appropriately indicated in Results.

**Ethics statement.** Mouse studies were performed in accordance with the recommendations in the Guide for the Care and Use of Laboratory Animals of the National Institutes of Health. The protocols were approved by the Institutional Animal Care and Use Committee (IACUC) at the Washington University School of Medicine (Assurance Number A3381-01) and the IACUC at the University of Texas Medical Branch (UTMB; Protocol Number 0209068B). Dissections and footpad injections were performed under anesthesia that was induced and maintained with ketamine hydrochloride and xylazine at the Washington University or isoflurane at UTMB. All efforts were made to minimize animal suffering. RMs experiments were reviewed and approved by the Vaccine Research Center Animal Care and Use Committee at the National Institute of Allergy and Infectious Diseases, the National Institutes of Health. The NHP experiments were performed in compliance with the pertinent regulations and policies from the National Institutes of Health.

**Viruses and cells.** The ZIKV Cambodian strain FSS13025 (GenBank number KU955593.1) was produced from an infectious cDNA clone[35]. The ZIKV-3′UTR-Δ10-LAV and ZIKV-3′UTR-Δ20-LAV strains were generated as described. The Zika Puerto Rico strain PRVABC59 (GenBank number KU501215) and Dakar 41519 strain (GenBank number HQ234501.1) were obtained originally from Dr Robert Tesh from the World Reference Center of Emerging Virus and Arboviruses (WRCEVA) at UTMB. The mouse-adapted ZIKV-Dakar 41519 strain was passaged twice in $Rag1^{-/-}$ mice (Jackson Laboratories) and as described previously[29]. Vero cells were purchased from the American Type Culture Collection (CCL-81; Bethesda, MD, USA), and maintained at 37 °C with 5% $CO_2$ in a high glucose Dulbecco modified Eagle medium (DMEM; Invitrogen, Carlsbad, CA, USA) with 10% fetal bovine serum (HyClone Laboratories, Logan, UT, USA) and 1% penicillin/streptomycin (Invitrogen). All cell lines tested negative for mycoplasma.

**Antibodies.** The following antibodies were used in this study: anti-mouse Ifnar1 monoclonal antibody (clone MAR1-5A3; Leinco Technologies, Inc., St. Louis, MO, USA); ZIKV-specific hyper-immune ascites fluids (obtained from WRCEVA); anti-mouse IgG antibody labeled with horseradish peroxidase (KPL, Gaithersburg, MD, USA); and goat anti-mouse IgG conjugated with Alexa Fluor 488 (Thermo Fisher Scientific, Providence, RI, USA).

**A129 mouse experiment.** A129 mice were bred in the animal facilities at UTMB. All mice were housed in pathogen-free mouse facilities. Three-week- or 15-week-old male A129 mice were infected with PBS (two sham groups), $10^4$ FFU of ZIKV-3′UTR-Δ10-LAV (Δ10), or $10^3$ FFU of ZIKV-3′UTR-Δ20-LAV (Δ20). Mice were anesthetized and bled via the retro orbital sinus for viremia testing. At day 28 post immunization, mice were measured for neutralizing antibody titers using an mCherry ZIKV infection assay[19]. On the same day, one sham group of PBS-immunized mice and Δ10- and Δ20-immunized mice were challenge with $10^6$ FFU of ZIKV PRVABC59. Another sham group of mice was used as an unchallenged negative control. At day 49 post immunization, mice were euthanized and necropsied. Epididymis and testes were harvested immediately as previously described[36]. Motile and nonmotile sperms were counted manually on a hemocytometer by microscopy. Total sperm counts equal to the sum of motile and nonmotile sperms. For quantification of viral loads, testes were homogenized and infectious viral levels were measured by a focus forming assay or qRT-PCR[19]. The qRT-PCT primer/probe set includes forward primer (1193F: 5′-CCGCTGCCCAACACAAG-3′), reverse primer (1269R: 5′-CCAC-TAACGTTCTTTTGCAGACAT-3′), and probe (5′-FAM/AGCCTACCT/ZEN/ TGACAAGCAATCAGACACTCAA/3IABkFQ-3′). The probe contains a 5′-FAM reporter dye, 3′ IBFQ quencher, and an internal ZEN quencher.

**Mouse pregnancy experiment.** C57BL/6J mice were bred and housed in pathogen-free mouse facilities at Washington University School of Medicine. One day prior to immunization, the 8-week-old female C57BL/6J mice were dosed with 0.5 mg of anti-Ifnar1 antibody via an intraperitoneal route. Subsequently, mice were subcutaneously inoculated in the footpad with $10^5$ FFU of ZIKV-3′UTR-Δ10-LAV or PBS sham. Immunized WT C57BL/6 female mice were mated with naive WT male mice. At E5, pregnant dams were injected intraperitoneally with 2 mg of anti-Ifnar1 antibody. On E6, mice were inoculated subcutaneously with $10^5$ FFU of mouse-adapted ZIKV Dakar 41519 via footpad injection. All animals were sacrificed on E13 and analyzed for viral loads in placentas, fetuses, and maternal tissues. Briefly, maternal blood, organs from dams (brain and spleen), and fetuses (placenta and fetal head) were collected. Serum was prepared after coagulation and centrifugation. Organs were weighed and homogenized using a bead-beater apparatus (MagNA Lyser, Roche). Viral RNA was extracted from serum and tissue samples using the RNeasy Mini kit (Qiagen). The viral RNA levels were determined by

TaqMan one-step qRT-PCR on an ABI 7500 Fast Instrument using standard cycling conditions. Viral burden is expressed on a log10 scale as viral RNA equivalents per gram or per milliliter after comparison with a standard curve produced using serial fivefold dilutions of ZIKV RNA from known quantities of infectious virus. The following primer/probe set was used for ZIKV qRT-PCR: forward primer (1183F: 5′-CCACCAATGTTCTCTTGCAGACATATTG-3′), reverse primer (1268R: 5′-TTCGGACAGCCGTTGTCCAACACAAG-3′), and probe (1213F: 5′-56-FAM/AGCCTACCT TGACAAGCAGTC/3IABkFQ-3′). Wherever indicated, viral burden for some samples was determined by focus-forming assay on Vero cells[20].

**Quantification of viral load in organs from A129 mice.** A129 mice were infected and organs were quantified for viral load using a focus-forming assay[20]. Viral RNA in testis also was quantified by qRT-PCR. Briefly, testes were harvested and placed in DMEM with beads for homogenization. After homogenization, the supernatant was used to extract viral RNA using RNeasy Mini kit (Qiagen). Extracted RNA was eluted in 40 μl RNase-free water. qRT-PCR assays were performed on the Light-Cycler 480 System (Roche) following the manufacturer's protocol by using a 50 μl reaction of the QuantiTect Probe RT-PCR Kit (Qiagen) and 10 μl RNA template. The viral load was calculated based on a standard curve produced using serial 10-fold dilutions of ZIKV RNA from known quantities of infectious virus. The qRT-PCT primer/probe set includes forward primer (1193F: 5′-CCGCTGCCCAACA-CAAG-3′), reverse primer (1269R: 5′-CCACTAACGTTCTTTTGCAGACAT-3′), and probe (5′-FAM/AGCCTACCT/ZEN/TGACAAGCAATCAGACACTCAA/ 3IABkFQ-3′). The probe contains a 5′-FAM reporter dye, 3′ IBFQ quencher, and an internal ZEN quencher.

**Vaccination of NHPs.** RM (*Macaca mulatta*) experiments were performed at Bioqual, Inc. (Rockville, MD, USA). All animal experiments were reviewed and approved by the Animal Care and Use Committee of the Vaccine Research Center, the National Institute of Allergy and Infectious Diseases, the National Institutes of Health. Animals were housed and cared in accordance with local, state, federal, and institutional policies in an American Association for Accreditation of Laboratory Animal Care-accredited facility at the Bioqual Inc. RMs (3–4/group) were randomized by body weight, gender, and age, and subcutaneously administered with $10^3$ FFU of parental WT ZIKV strain FSS13025, ZIKV-3′UTR-Δ10-LAV, ZIKV-3′ UTR-Δ20-LAV, or PBS sham at day 0. Blood was collected at days 2, 3, 4, 5, 7, and 10 for viremia testing and weekly for analysis of antibody responses by an mCherry ZIKV neutralization assay[19, 37]. The immunized animals were subcutaneously challenged with $10^3$ FFU of ZIKV strain PRVABC59 at week 8. Blood samples were collected for determination of viral load at days 2, 3, 4, 5, 7, and 10, and neutralizing antibody at weeks 2, 4, and 6 post challenge.

Viremia from RMs was quantified by a qRT-PCR assay and focus-forming assay as described in the proceeding section. For qRT-PCR quantification, viral RNA was extracted from rhesus serum using QIAamp Viral RNA Kits (Qiagen) following the manufacturer's instruction. Extracted RNA was eluted in 40 μl RNase-free water. qRT-PCR assay was performed as described above. In vitro-transcribed full-length ZIKV RNA was used as a standard for qRT-PCR quantification. The primer/probe set is described in the proceeding section.

**Neutralization assay and neurovirulance.** All antibody neutralization titers were determined using an mCherry ZIKV as previously reported[19]. The serum dilution that neutralized 50% of mCherry ZIKV infection ($NT_{50}$) were presented. For measuring neurovirulence, 1-day-old outbred CD-1 mice (Charles River) were injected intracranially with indicated amounts of viruses. The infected mice were monitored for morbidity and mortality as reported previously[19].

**Mosquito infection.** For measuring mosquito infection, artificial blood-meal spiked with $10^6$ FFU/ml of indicated viruses was used to feed *A. aegypti* mosquitoes (derived from a stock isolated in Galveston, TX, USA), and engorged mosquitoes were incubated at 28 °C, 80% relative humidity on a 12:12 h light:dark cycle with ad libitum access to 10% sucrose. The infection rates were determined at day 7 post feeding as reported previously[35].

**Data analysis.** All data were analyzed with GraphPad Prism v7.02 software. Data are expressed as the mean ± SD. Comparisons of groups were performed using Mann–Whitney test or one-way analysis of variance with a multiple comparisons correction. A $P$-value of < 0.05 indicates statistical significance.

**Data availability.** The authors declare that the data supporting the findings of this study are available within the article and its Supplementary Information files, or are available from the authors upon request.

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

## Acknowledgements

We thank colleagues at University of Texas Medical Branch for helpful discussions and support during the course of this study. We also thank J.P. Todd and R. Bailer for help with the nonhuman primate studies and sample processing. P.-Y.S. lab was supported by University of Texas Medical Branch (UTMB) startup award, University of Texas STARs Award, CDC grant for the Western Gulf Center of Excellence for Vector-Borne Diseases, Pan American Health Organization grant SCON2016-01353, the Kleberg Foundation award, UTMB CTSA UL1TR-001439 and NIH grant AI127744. K.M.M., W.-P.K., B.S.G., and T.C.P. are funded through intramural funding from the National Institute of Allergy and Infectious Diseases. This research was also partially supported by NIH grant AI120942 to S.C.W. and grants AI073755, AI104972, and AI106695 from the NIH to M.S.D. P.F.C.V. was suported by projects of CAPES (Zika Fast-Track) and CNPq: grants 440405/2016-5, and 303999/2016-0 from the Ministry of Science and Technology of Brazil and by the Ministry of Health.

## Author contributions

C.S., A.E.M., B.W.J., J.R., B.T.D.N., D.B.A.M., and X.X. performed experiments and data analysis. K.M.M., W.-P.K., T.C.P., A.D.B., S.C.W., S.L.R., P.F.C.V., B.S.G., M.S.D., and P.-Y.S. designed the experiments and interpreted the results. C.S., K.M.M., T.C.P., A.D.B., S.C.W., S.L.R., P.F.C.V., B.S.G., M.S.D., and P.-Y.S. wrote the manuscript.

## Additional information

**Competing interests:** C.S. and P.-Y.S. have filed a patent related to the technology presented in this paper. M.S.D. is a consultant for Inbios, Visterra, and Takeda Pharmaceuticals and on the Scientific Advisory Boards of Moderna and OvaGene. The remaining authors declare no competing financial interests.

