## [Peer Review File · Nature Communications]

REVIEWERS' COMMENTS:

Reviewer #2 (Remarks to the Author):

The revised manuscript addresses the major and minor points from my initial review. My one minor request is that the NHP data added as Figure S4 should be included as one or more panels in a main display item as opposed to supplemental data (even though the data is negative and not that graphically appealing, it helps to draw a contrast between the two LAV constructs). The paragraph describing these results should also be edited for clarity and word choice (e.g., 'illicit' should be 'elicit')

--Dave O'Connor

Point-by-point response to reviewer's comments

Reviewer #2 (Remarks to the Author):

The revised manuscript addresses the major and minor points from my initial review. My one minor request is that the NHP data added as Figure S4 should be included as one or more panels in a main display item as opposed to supplemental data (even though the data is negative and not that graphically appealing, it helps to draw a contrast between the two LAV constructs). The paragraph describing these results should also be edited for clarity and word choice (e.g., 'illicit' should be 'elicit')

Response: Done. We have integrated the supplemental data in Fig. S4 to the main display Fig. 3. We have also corrected the typo and improved the description of this result.